# The Lipidomic Profile Is Associated with the Dietary Pattern in Subjects with and without Diabetes Mellitus from a Mediterranean Area

**DOI:** 10.3390/nu16121805

**Published:** 2024-06-08

**Authors:** Marina Idalia Rojo-López, Maria Barranco-Altirriba, Joana Rossell, Maria Antentas, Esmeralda Castelblanco, Oscar Yanes, Ralf J. M. Weber, Gavin R. Lloyd, Catherine Winder, Warwick B. Dunn, Josep Julve, Minerva Granado-Casas, Dídac Mauricio

**Affiliations:** 1Institut de Recerca Sant Pau (IR SANT PAU), Sant Quintí 77-79, 08041 Barcelona, Spain; nut.marina.rojo.l@gmail.com (M.I.R.-L.); barmaria95@gmail.com (M.B.-A.); jrossell@santpau.cat (J.R.); maantentas@gmail.com (M.A.); jjulve@santpau.cat (J.J.); 2B2SLab, Departament d’Enginyeria de Sistemes, Automàtica i Informàtica Industrial, Universitat Politècnica de Catalunya, 08028 Barcelona, Spain; 3Networking Biomedical Research Centre in the Subject Area of Bioengineering, Biomaterials and Nanomedicine (CIBER-BBN), 28029 Barcelona, Spain; 4Centro de Investigación Biomédica en Red de Diabetes y Enfermedades Metabólicas Asociadas, Instituto de Salud Carlos III, 28029 Madrid, Spain; oscar.yanes@urv.cat; 5Department of Internal Medicine, Endocrinology, Metabolism and Lipid Research Division, Washington University School of Medicine, St. Louis, MO 63110, USA; esmeraldacas@gmail.com; 6Department of Electronic Engineering, Institut d’Investigació Sanitària Pere Virgili (IISPV), Universitat Rovira i Virgili, 43007 Tarragona, Spain; 7School of Biosciences, University of Birmingham, Edgbaston, Birmingham B15 2TT, UK; r.j.weber@bham.ac.uk (R.J.M.W.); g.r.lloyd@bham.ac.uk (G.R.L.); c.l.winder@liverpool.ac.uk (C.W.); warwick.dunn@liverpool.ac.uk (W.B.D.); 8Phenome Centre Birmingham, University of Birmingham, Edgbaston, Birmingham B15 2TT, UK; 9Institute of Metabolism and Systems Research, University of Birmingham, Edgbaston, Birmingham B15 2TT, UK; 10Centre for Metabolomics Research, Department of Biochemistry, Cell and Systems Biology, Institute of Systems, Molecular and Integrative Biology, University of Liverpool, Liverpool L69 7ZB, UK; 11Department of Nursing and Physiotherapy, University of Lleida, 25198 Lleida, Spain; 12Research Group of Health Care (GreCS), IRBLleida, 25198 Lleida, Spain; 13Department of Endocrinology and Nutrition, Hospital de la Santa Creu i Sant Pau, 08041 Barcelona, Spain; 14Faculty of Medicine, University of Vic (UVIC/UCC), 08500 Vic, Spain

**Keywords:** lipidomic, diabetes, dietary pattern, lipid species

## Abstract

Lipid functions can be influenced by genetics, age, disease states, and lifestyle factors, particularly dietary patterns, which are crucial in diabetes management. Lipidomics is an expanding field involving the comprehensive exploration of lipids from biological samples. In this cross-sectional study, 396 participants from a Mediterranean region, including individuals with type 1 diabetes (T1D), type 2 diabetes (T2D), and non-diabetic individuals, underwent lipidomic profiling and dietary assessment. Participants completed validated food frequency questionnaires, and lipid analysis was conducted using ultra-high-performance liquid chromatography coupled with mass spectrometry (UHPLC/MS). Multiple linear regression models were used to determine the association between lipid features and dietary patterns. Across all subjects, acylcarnitines (AcCa) and triglycerides (TG) displayed negative associations with the alternate Healthy Eating Index (aHEI), indicating a link between lipidomic profiles and dietary habits. Various lipid species (LS) showed positive and negative associations with dietary carbohydrates, fats, and proteins. Notably, in the interaction analysis between diabetes and the aHEI, we found some lysophosphatidylcholines (LPC) that showed a similar direction with respect to aHEI in non-diabetic individuals and T2D subjects, while an opposite direction was observed in T1D subjects. The study highlights the significant association between lipidomic profiles and dietary habits in people with and without diabetes, particularly emphasizing the role of healthy dietary choices, as reflected by the aHEI, in modulating lipid concentrations. These findings underscore the importance of dietary interventions to improve metabolic health outcomes, especially in the context of diabetes management.

## 1. Introduction

Lipids play essential roles in human life such as in energy storage and as precursors for metabolic processes and structural functions [1]. The lipidome, the total content of lipids in biological samples, consists of around 200,000 different molecular species with diverse abundance [2]. Lipidomics is a branch of metabolomics that focuses on the comprehensive investigation of lipid species (LS), their quantities, biological functions, as well as their subcellular localization and distribution across cells, tissues, and bodily fluids [3]. Both endogenous and exogenous factors, such as genetics, age, medication, disease, and lifestyle, including dietary patterns, play a key role in shaping the dynamics of the lipidome [1]. Dietary patterns are connected to circulating metabolites, given that certain lipids must be sourced from the diet as they cannot be produced within the body, leading to a significant influence on metabolism [3,4].

The lipidomic profile of subjects with prediabetes, type 1 diabetes (T1D), and type 2 diabetes (T2D) has been previously analyzed, and differences have been reported compared to healthy subjects [5,6,7,8,9,10,11,12,13]. However, few studies have demonstrated the impact of dietary patterns on the lipidomic profile in subjects with diabetes. In a double-blind, randomized trial, individuals with T2D who consumed a Western diet enriched with medium-chain fatty acids (MCFAs) showed reduced levels of specific plasma sphingolipids (SP), ceramides (Cer), and acyl-carnitines (AcCa), compared to those on a standard Western diet high in long-chain fatty acids (LCFAs) [14]. Additionally, improvements in fasting insulin levels were correlated with changes in certain SP [14]. Furthermore, in an intervention study in which participants were randomly assigned to a 12-week Paleolithic-type diet with (diet-ex) or without supervised exercise (diet), the diet group exhibited an overall reduction in Cers, diacylglycerols (DAG), and various phosphatidylcholines (PC), sphingomyelins (SM), and triacylglycerols (TAG) [15]. In addition, the diet-ex group demonstrated decreases in specific lipids and increases in others, such as DAGs, PCs, and TAGs [15]. Both diet and diet-ex groups showed decreased polyunsaturated fatty acid (PUFA)-TAGs [15]. In a pilot study of the Diabetes Interventional Assessment of Slimming or Training tO Lessen Inconspicuous Cardiovascular Dysfunction (DIASTOLIC) trial, individuals with T2D participated in a 12-week low-energy (~810 kcal/day) meal-replacement plan (MRP) and were compared to healthy controls (HC) [16]. Within the T2D group, 12 LS, comprising eight SPs and four Cers, were initially downregulated [16]. However, after the 12-week MRP, their levels approached those of HC [16]. In addition, a PREDIMED substudy, which included individuals with T2D, revealed that those following a Mediterranean diet (MDiet) supplemented with nuts induced modest changes in the lipid profile [17]. Significant alterations in cholesteryl ester concentrations were observed in the MDiet with nuts group compared to the control group [17]. Additionally, the MDiet supplemented with olive oil showed greater increases in lipids with a longer mean acyl chain length compared to the control group, contrasting with lipids with a shorter acyl chain length [17]. Likewise, in a three-arm randomized trial (Mediterranean diet, traditional Chinese diet, and a control group) with individuals at risk of T2D, the Mediterranean and traditional Chinese diets led to distinct lipidomic changes compared to the control group, including alterations in TAG fractions, alkyl phosphatidylethanolamines and alkenyl phosphatidylethanolamines [18]. Moreover, dietary intervention in subjects with metabolic syndrome employing a Nordic Diet found lower insulin resistance-inducing Cer levels and higher antioxidative plasmalogens (phospholipids) after 18 or 24 weeks; however, most changes were transient [19]. In a 12-week controlled dietary intervention for individuals with impaired glucose metabolism, participants were assigned to a healthy diet (whole grains, fatty fish, and bilberries), a whole-grain enriched diet, and a control diet [20]. The study identified significant lipid changes, highlighting increased levels of specific TGs and lysophosphatidylcholines (LPC) in the healthy diet group [20]. Additionally, a crossover randomized trial was performed with individuals with T1D following a low-carbohydrate diet, revealing an elevation in SMs and PCs [21]. Furthermore, in a crossover, controlled feeding study involving healthy subjects receiving low or high-glycemic load diets, a relative shift in LS was observed; however, the overall pool did not exhibit changes [22].

In the present study, we hypothesized that dietary patterns would correlate with variations in plasma lipid concentrations among individuals with T1D, T2D, and non-diabetic individuals. Thus, the aim of the study was to analyze the association between lipidomic profile and eating habits using the alternate Mediterranean Diet score (aMED) and the alternate Healthy Eating Index (aHEI) [23,24] in a population of T1D and T2D, and non-diabetic subjects from a Mediterranean area. The aMED and aHEI are two widely used dietary quality indexes to assess the Mediterranean Diet and healthy eating in individuals with diabetes. Both scores have shown a potential relationship with the risk of developing T2D and cardiovascular diseases [24,25]. For this reason, the aMED and aHEI are well fitted to assess the dietary pattern of a Mediterranean population with diabetes. The interaction between the dietary pattern and diabetes condition was also investigated.

## 2. Materials and Methods

### 2.1. Study Design

This was a cross-sectional study. A sample of 396 participants was recruited from three different cohorts from northeastern Spain (Lleida and Barcelona): 119 individuals diagnosed with T1D, 88 subjects with T2D, and 189 individuals without diabetes. Detailed descriptions of the study cohorts, originating from prior studies conducted at University Hospitals Arnau de Vilanova (Lleida, Spain), Germans Trias i Pujol (Badalona, Spain), Clinic (Barcelona, Spain), and the Mollerussa Primary Care Center (Lleida, Spain), can be found in previous publications [26,27,28]. Briefly, the inclusion criteria were as follows: for individuals with T1D, a diagnosis lasting more than 1 year and age > 18 years; for individuals with T2D, a diagnosis of T2D and age 40–75 years; individuals without diabetes were selected according to the American Diabetes Association criteria (values of HbAc1 < 6.5% and fasting plasma glucose < 126 mg/dL) and age > 25 years. Individuals without diabetes were recruited from the Germans Trias i Pujol (Badalona, Spain) and the Primary Care Center Mollerussa (Lleida, Spain) cohorts. These were individuals without diabetes who voluntarily agreed to participate in the study. Exclusion criteria for all participants included being a healthcare professional, showing signs of mental disorders, having a history of previous cardiovascular disease or diabetic foot disease, chronic kidney disease (defined as an estimated glomerular filtration rate < 60 mL/min or a urine albumin/creatinine ratio over 300 mg/g), pregnancy, and other conditions that could influence the study results; moreover, in the control group, individuals with other types of diagnosed diabetes were also excluded. The local Ethics Committee from University Hospital Germans Trias i Pujol approved the study (PI-15-147). Written informed consent was obtained from all the participants.

### 2.2. Clinical Data

Blood samples were collected to determine biochemical variables using standard lab procedures. Furthermore, clinical and sociodemographic data were collected from anamnesis and medical records. Anthropometric variables (i.e., weight, height, and body mass index [BMI]) were obtained by standardized methods [27,28]. The use of any antihypertensive or lipid-lowering drugs was used to define the presence of hypertension and dyslipidemia, respectively. Regular physical activity was defined as performing for more than 25 min/day of any physical activity which expends a minimum of 4 METS (metabolic equivalent of task), such as walking, and sedentarism was defined if the physical activity lasted less than 25 min/day according to Bernstein et al. [29] and Cabrera de León et al. [30]. Smoking habits were determined including former and current smokers.

### 2.3. Dietary Pattern Assessment

Diet was evaluated by the validated 101-item semiquantitative Food Frequency Questionnaire Consumption (FFQC) [31,32], which was administered via personal interviews by specialized and trained researchers. The FFQC collects data on habitual food consumption over the previous year’s visit. The dietary pattern was assessed using the aMED and the aHEI [23,24]. The aMED assesses adherence to the MDiet rating from 0 (minimum) to 9 (maximum) according to the overall consumption of whole grains, vegetables, fruits, legumes, nuts, fish, the ratio of monounsaturated fat (MUFA) to saturated fat (SFA), red and processed meats, and alcohol [23]. The aHEI assigns a score ranging from 0 (non-adherence) to 87.5 (healthy eating) according to the consumption of vegetables, fruit, whole grains, nuts, legumes, and vegetable protein, white and red meats, trans fat, PUFA and SFA, and alcohol intake [24]. Composition food tables from the US Department of Agriculture, and Spanish and English food sources were used to estimate nutrient intake [33,34,35]; the nutrient intake was adjusted for energy intake.

### 2.4. Lipidomic Study

For the extraction of lipids, frozen serum samples (50 µL) were thawed on ice, and aliquots of each sample were taken to create a pooled quality control (QC) that represented all the samples in the study. The pooled QC was thoroughly mixed, and aliquots (50 µL) were prepared for analyses in each of the 6 sample batches. Both the QCs and biological samples were stored at −80 °C until each batch’s analysis. To extract the lipids from the samples, 50 µL of serum from either the biological sample or QC was combined with 150 µL of isopropanol (LC-MS grade), vortexed for 20 s, and centrifuged at 22,000× *g*, 4 °C for 20 min. Next, 150 µL of the resulting supernatant was transferred to a vial with low recovery capabilities and subsequently moved to the LC sample manager at 4 °C.

Ultra-High-Performance Liquid Chromatography-Mass Spectrometry (UHP-LC/MS) analysis was conducted using a Dionex UltiMate 3000 Rapid Separation LC system (Thermo Fisher Scientific, Waltham, MA, USA), coupled with a heated electrospray Q Exactive Focus mass spectrometer (Thermo Fisher Scientific, Waltham, MA, USA). Non-polar extracts were reconstituted using a mixture of isopropanol and water (75:25) and subjected to analysis on a Hypersil GOLD column (100 × 2.1 mm, 1.9 µm; Thermo Fisher Scientific, Waltham, MA, USA). Mobile phase A was comprised of a solution containing 10 mM ammonium formate and 0.1% formic acid in a mixture of 60% acetonitrile and water, whereas mobile phase B consisted of 10 mM ammonium formate and 0.1% formic acid dissolved in a combination of 90% propan-2-ol and water. The flow rate was set to 0.40 mL/min, and a gradient program was applied, starting at *t* = 0.0 with 20% B, remaining at 20% B at *t* = 0.5, increasing to 100% B at *t* = 8.5, holding at 100% B at *t* = 9.5, and then returning to 20% B at *t* = 11.5, maintaining this composition until *t* = 14.0. All changes followed a linear pattern with a curve setting of 5. The column temperature was maintained at 55 °C, and the injection volume was 2 μL. Data acquisition was performed separately in positive and negative ionization modes over the mass range of 150–2000 mass-to-charge (*m*/*z*) at a resolution of 70,000 (FWHM at *m*/*z* 200). The ion source parameters were set as follows: sheath gas at 50 arbitrary units, auxiliary gas at 13 arbitrary units, sweep gas at 3 arbitrary units, spray voltage at 3.5 kV for positive ion mode and 3.1 kV for negative ion mode, capillary temperature at 263 °C, and auxiliary gas heater at 425 °C. Data-dependent MS2 in ‘Discover mode’ was utilized for MS/MS spectral acquisition with the following settings: resolution at 17,500 (FWHM at *m*/*z* 200), isolation width of 3.0 *m*/*z*, and stepped normalized collision energy at 20%, 50%, and 80%. Spectra were collected across three mass ranges: 200–400 *m*/*z*, 400–700 *m*/*z*, and 700–1500 *m*/*z* on the pooled QC samples [36]. The instrument control software utilized was Thermo ExactiveTune (version 2.8 SP1 build 2806) for both cases, and data were acquired in profile mode. QC samples were acquired in both profile and dependent scan modes at the beginning of the run (i.e., 7 QCs MS1 only, 3 QCs with MS2) and then repeated every seventh injection with two QC samples at the end of the analytical batch [36]. Preparation blank samples were analyzed between QCs 5 and 6 and at the end of the analytical batch [36].

### 2.5. Data Analysis

The raw data from each analytical batch were converted from the instrument-specific format to the mzML file format using the open-access ProteoWizard msconvert tool (version 3.0.11417) [37]. Deconvolution was conducted with the R package XCMS (version 1.46.0, running in the Galaxy workflow environment) [38]. The XCMS peak picking parameters were optimized using Isotopologue Parameter Optimization (IPO—version 1.0.0) [39]. A data matrix of metabolite features (*m*/*z*-retention time pairs) versus samples was created with provided peak areas [36]. The initial five QCs from each batch were used to stabilize the analytical system and were subsequently excluded before processing and analysis [36]. Intensity drift was corrected for each lipid feature using the Quality Control-Robust Spline Correction algorithm [40] in the pmp R package (version 1.6.0) [41]. Principal Component Analysis (PCA) was employed to detect and remove suspected outlier (QC) samples within each batch to ensure robust correction (using PCs 1 and 2, Hotelling T^2^*p* < 0.05) [36]. Blank samples at the beginning and end of each run were used to exclude non-biological features [36]. Features with an average QC intensity less than 20 times the average intensity of the blanks were removed [36]. The 80% rule was applied to remove features containing missing values in more than 20% of the samples, and features with RSD > 30% and present in less than 90% of the QC samples were also discarded [36]. The remaining missing values were imputed using the minimum value of the feature divided by two [36]. Lipid annotation was conducted using LipidSearch (version 4.2.21), and confirmation was defined via annotations graded as A or B, which were linked to XCMS-detected features exhibiting an absolute ppm error of less than 5 and an absolute retention time deviation of less than 5 s.

### 2.6. Statistical Analysis

To perform the descriptive analysis between the different study groups (T1D, T2D, and non-diabetic as control), the compareGroups R package (version 4.8.0) [42] was employed to summarize participants’ continuous variables as mean (standard deviation) and categorical data as frequency (percentage) in the clinical dataset with a minimum level of statistical significance set at *p* < 0.05 for each comparison. Simultaneously, UHPLC-MS features underwent log transformation, scaling, and centering, followed by the application of linear regression models to each specific feature. The linear regression models were adjusted by sex, age, hypertension, dyslipidemia, BMI, waist circumference, glycated hemoglobin (HbA1c), glucose, smoking, and physical activity; diabetes condition was used when analyzing all subjects; diabetes duration was used when analyzing T1D and T2D. All numeric variables of interest, as well as the variables used for adjustment, were scaled to unit variance and mean centered. Diabetes treatment was included in the analysis of subjects with T2D stratified into four categories: oral Antidiabetic (OAD), OAD + insulin, insulin, and other. Subjects with T1D did not receive any glucose-lowering drug other than insulin. In all analyses, False Discovery Rate (FDR) correction was applied, and a corrected *p*-value of <0.05 was deemed significant. Another aspect of the analysis involved determining the association between each LS and the interaction between the diabetes condition and the dietary index. Furthermore, the association between each LS and the consumption of carbohydrates, fats, and protein was also analyzed.

## 3. Results

### 3.1. Demographic and Clinical Characteristics

Demographic and clinical characteristics of the study participants are provided in Table 1. Briefly, individuals with T2D were older (mean (SD) = 58.3 (10.1); *p* < 0.001) and had a higher BMI (mean (SD) = 31.7 (5.6); *p* < 0.001), as well as a larger waist circumference (mean (SD) = 106 (12.3); *p* < 0.001), and higher HbA1c levels (mean (SD) = 8.1 (1.5); *p* < 0.001). The T2D group had a higher proportion of individuals with hypertension (56.8%; *p* < 0.001) and dyslipidemia (56.8%; *p* < 0.001) and were more sedentary (54%; *p* < 0.001) compared to individuals with T1D and controls.

Regarding diet (Table 2), individuals with T2D showed higher protein (*p* < 0.001), carbohydrate (*p* < 0.001), and omega-3 consumption (*p* = 0.002) than individuals with T1D and controls. Additionally, individuals with T1D showed higher total fat consumption (*p* < 0.001), including SFA (*p* < 0.001), MUFA (*p* < 0.001), PUFA (*p* = 0.003), and omega-6 fatty acids (*p* = 0.003). Furthermore, individuals with T2D showed higher aMED and aHEI scores (*p* < 0.001, and *p* < 0.001, respectively) (Table 2 and Figure 1).

### 3.2. Lipids Associated with Dietary Pattern

In the lipidomic analysis across all subjects, a total of 66 features were significantly associated with the aHEI index, 56 in positive ionization mode and 10 in negative ionization mode. However, only two of these features, which showed a negative association with the aHEI and were acquired in positive acquisition mode, could be annotated. Additionally, one unannotated feature was associated with T1D.

When considering control subjects alone, a significant association was found between 57 lipidic features and the aHEI index. Among these, five features obtained a confirmed annotation, and two showed a negative association in the positive ionization mode; in the negative ionization mode, three features were confirmed, with one having a positive association and two having a negative association with the aHEI. There were no lipidic features associated with the aHEI in T2D. Data are presented in Figure 2, Appendix A, and Table 3, indicating the FDR corrected *q*-value for the analysis across the study groups. Moreover, no features were associated with the aMED index in either group.

### 3.3. Lipids and Macronutrient Intake

Regarding the association between dietary carbohydrates and lipidomics in positive ionization mode using all subjects, an association with 51 lipidic features was found; 4 of them obtained a confirmed annotation with a positive association (Figure 2, Appendix A, and Table 3). In negative ionization mode using all subjects, 38 lipidic features were observed, from which only 2 of them had a confirmed annotation and showed a positive association. In the analysis performed in non-diabetic subjects, 1 of 25 lipidic features obtained a confirmed annotation with a negative association.

The association between lipids and dietary fat was also analyzed. In the analysis involving all subjects in both positive and negative ionization mode, 72 lipid features were observed, with 3 of them having a confirmed annotation, 1 of them with a positive association, and 2 with a negative association (Figure 2, Appendix A, and Table 3). No associations were observed with T1D and T2D. Regarding non-diabetic subjects, 37 features showed an association with dietary fat, but none of them were annotated.

The correlation between lipids and dietary protein was further examined. In the comprehensive analysis encompassing T2D subjects in positive ionization mode, 289 lipid features were detected, and 12 of them were successfully annotated (Figure 2, Appendix A, and Table 3), with three of them confirmed to be positively associated and two of them negatively associated. No significant associations were found in any other group.

### 3.4. Interaction between Diabetes and the Dietary Pattern

To further investigate the interaction between the condition diabetes and the aHEI index in the positive ionization mode, four lipids were plotted to visualize their relationship with the dietary index in each group (Figure 2). LPC(18:2e) exhibited a positive association with the aHEI index in both non-diabetic and T2D subjects, suggesting a higher presence of LPC(18:2e) with a higher aHEI index. However, the opposite was observed in T1D subjects, where the clear negative association provides an explanation for the significant interaction between diabetes condition and the aHEI index (beta = −0.398 and *q*-value = 0.00541). Interestingly, T1D subjects with high aHEI indices had a similar LPC(18:2e) value as that of non-diabetic or T2D subjects. A similar pattern occurred with both LPC(18:1e) and PC(40:3) with an interaction of beta = −0.429 and *q*-value = 0.00216 and beta = −0.539; *q*-value = 0.00348, respectively; meanwhile, PC(32:1) showed the opposite pattern with an interaction of beta = 0.420 and *q*-value = 0.03365 (Figure 3 and Appendix A). Only LPC(18:2e) and LPC(18:1e) obtained a confirmed annotation. In the negative ionization mode, no feature reached statistical significance.

Subsequently, the association between lipids and aMED and diabetes condition was further analyzed. In the case of the interaction with T1D, 18 features were found to be statistically significant in the positive ionization mode; however, none of these features were annotated. Regarding the interaction with T2D in the positive ionization mode, no feature achieved statistical significance. On the other hand, the negative ionization mode did not reveal any significant features.

## 4. Discussion

In the current study, we assessed concentrations of serum LS among individuals who were non-diabetic or had either T1D or T2D and investigated the association between lipidomic information and dietary habits. Furthermore, we explored the interaction between dietary patterns and diabetes. We annotated several lipids associated with the aHEI, including AcCa(14:0) and TG(52:1), which showed a negative association across all subjects. Moreover, AcCa(14:0), ChE(16:1), PC(32:1), and PI(32:1) were negatively associated with the non-diabetic group, while PC(40:7) showed a positive association. However, we did not find any significant correlation between lipid features and the aMED in any of the study groups. Regarding specific components of the diet, carbohydrates showed a positive association with PI(36:4) and PCs in all subjects, whereas TGs showed a negative association with carbohydrates in all subjects, and Cer(t42:1) showed a positive association in non-diabetic controls. Representative lipids associated with fats in all subjects were negatively associated with PCs and positively associated with SM(d37:1). Interestingly, in the analysis of dietary proteins, we observed associations exclusively in T2D subjects, with AcCas being negatively associated and PC(38:5), SM(d42:2), and TG(54:4) positively associated. Therefore, in the interaction analysis between diabetes and the aHEI, we found a similar direction between non-diabetic controls and T2D subjects, while the opposite was observed in T1D subjects in LPC(18:2e) and LPC(18:1e). This directional shift may seem counterintuitive at first glance, given that individuals with T1D and T2D exhibited higher aHEI scores, indicating healthier dietary habits, compared to non-diabetic subjects. However, despite having better overall dietary quality, individuals with T1D displayed a different response in certain lipid characteristics compared to both controls and those with T2D. One potential explanation for this discrepancy could be the complex interplay between dietary factors, metabolic processes, and the unique physiological characteristics of each type of diabetes. While individuals with T1D and T2D may have adopted healthier eating habits, the underlying metabolic dysregulation associated with diabetes, particularly in T1D, could lead to divergent responses in lipid profiles despite similar dietary patterns. In our previous study, we annotated significant differences in the lipid profile between subjects with T1D and T2D [36]. Specifically, LPCs and ceramides exhibited opposite trends, with LPCs being elevated in T1D but reduced in T2D, while ceramide levels were elevated in T2D but reduced in T1D [36]. Additionally, PCs were notably decreased in individuals with T1D. In terms of glycemic status, there was a progressive increase observed in a cluster of 1-deoxyceramides from normoglycemia to prediabetes and ultimately to T2D [36]. The results suggest that a healthy diet might regulate metabolic dysfunctions associated with diabetes.

There is a limited body of studies analyzing the lipidomic profile and dietary patterns in subjects with diabetes. In a study investigating dietary impacts in relation to incident diabetes in a Chinese population, four significant glycerophospholipids (GPLs) [PC(16:0/16:1), PC(16:0/18:1), PC(18:0/16:1), PE(16:0/16:1)] correlated positively with carbohydrate intake and the carbohydrate/fat ratio, but negatively with fat intake [43]. These four GPLs, linked to de novo lipogenesis, exhibited associations with unhealthy dietary patterns characterized by high consumption of refined grains and low intake of fish, dairy, and soy products [43]. In our study, we observed an association between PC(32:1) and the aHEI among non-diabetic subjects. Furthermore, we annotated an association between PC(38:3) and carbohydrate and fat intake among all study participants. Additionally, we observed an association involving PC(38:5) with protein consumption, specifically within the subset of individuals with T2D.

A 12-week crossover trial among adults with T1D switching between low- and high-carbohydrate diets revealed changes in 11 LS [21]. The most notable changes were observed in LPC(O-16:0), monounsaturated SM(d36:1), and polyunsaturated PC(P-36:4)/PC(O-36:5), particularly increased after the low carbohydrate diet [21].

In the context of T2D, there are several reported associations. The Dietary Intervention and Vascular Function randomized controlled trial (DIVAS) investigated the impact of isoenergetic dietary fat modification [44]. Participants followed diets rich in SFAs (control), MUFA, or a combination of MUFA and *n*-6 PUFA for 16 weeks [44]. Phospholipids like PE(20:3) and PC(20:3), along with CE(20:3), showed significant positive associations with T2D [44]. The MUFA-rich diet notably increased concentrations of specific lipids such as TG(22:1), SM(24:1), and TG(18:2) while reducing others like DG(16:0), DG(18:0), TG(16:0), and TG(18:0) [44]. Similarly, a diet rich in mixed unsaturated fatty acids altered lipid concentrations by reducing certain DGs, PE, and SM species while elevating TGs, lactosylceramide, and CE(24:0) [44]. Moreover, elevated levels of PC(16:0/22:6) and PC(18:2/22:6) were observed in the MDiet group compared to other diets in a recent study involving Chinese subjects at risk of T2D (overweight and with prediabetes) [18]. We found a positive association between protein intake and PC(38:5) in T2D subjects. Interestingly, a study involving men homozygous for the TCF7L2 genotypes, which is associated with a high risk of T2D, revealed lower SP concentrations and reduced AcCa after normal carbohydrate meals, suggesting an intricate interplay between genetic predisposition and dietary carbohydrate content [45]. In the analysis of the carbohydrate content of the diet and lipidomics, we did not find any significantly different LS in T2D subjects.

This study has various strengths. Firstly, it incorporates a sizable sample size of 396 subjects, ensuring reliability in the statistical analysis. Secondly, the population under study contains non-diabetic subjects as well as subjects with T1D and T2D, which enables a comprehensive approach to the association between lipidomics and diet, considering diabetes. However, limitations exist. Validation of our findings in an independent cohort was not feasible, and the observational design of our study precludes the establishment of causal relationships.

## 5. Conclusions

This study provides valuable insights into the association between serum LS and dietary habits among individuals with T1D, T2D, and non-diabetic subjects. Our findings reveal significant associations between specific LS and dietary patterns, particularly with the aHEI and certain lipids, such as AcCa and TG, which showed consistent associations across all subjects. Interestingly, distinct lipid associations were observed among different diabetic conditions, including LPC with T1D and PC with the non-diabetic control group. However, our analysis did not find any significant association with the aMED score. Carbohydrates exhibited positive associations with several lipid types, while fats were predominantly linked with phosphatidylcholines and sphingomyelins. Further research is warranted to comprehensively assess lipidomic profiles in relation to glucose tolerance and dietary patterns.

## Figures and Tables

**Figure 1 nutrients-16-01805-f001:**
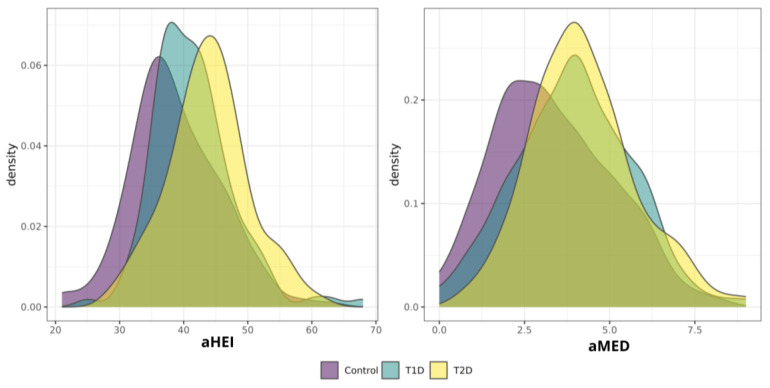
Density plot for aHEI and aMED scores. This density plot illustrates the distribution of two continuous variables, aHEI and aMED, within a population of non-diabetic control individuals, as well as T1D and T2D individuals. The *x*-axis represents the scores, while the *y*-axis represents the population density. aMED, alternate Mediterranean Diet score; aHEI, alternate Healthy Eating Index; T1D, type 1 diabetes; T2D, type 2 diabetes.

**Figure 2 nutrients-16-01805-f002:**
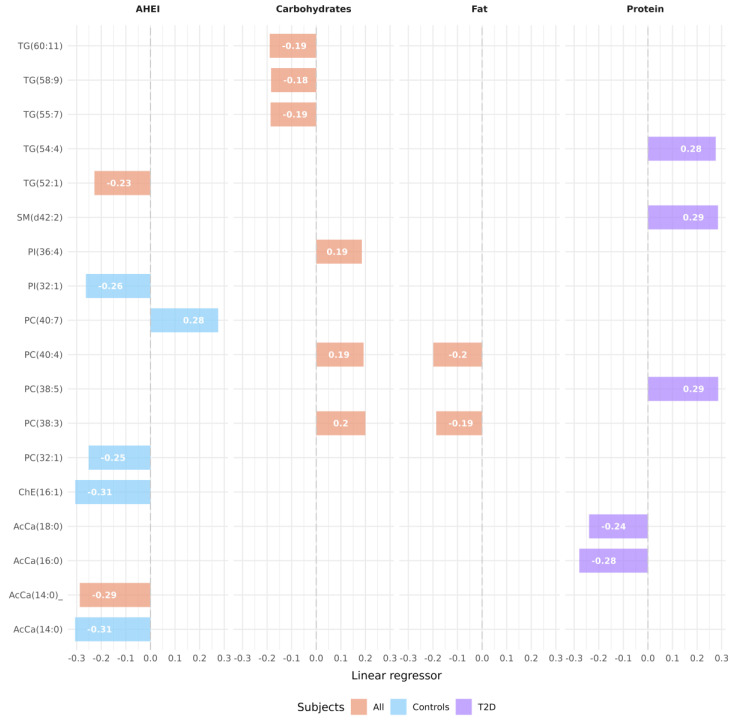
Bar plot of linear regressors of lipids significantly associated with the aHEI, carbohydrates, fats, and proteins in the studied subjects. aHEI, alternate Healthy Eating Index; All, all subjects; Controls, non-diabetic subject; T2D, type 2 diabetes. Each bar represents the regression coefficient (beta value) of a specific lipid species with respect to the corresponding dietary factor, providing insights into the relationships between lipid metabolism and dietary intake patterns in the study population.

**Figure 3 nutrients-16-01805-f003:**
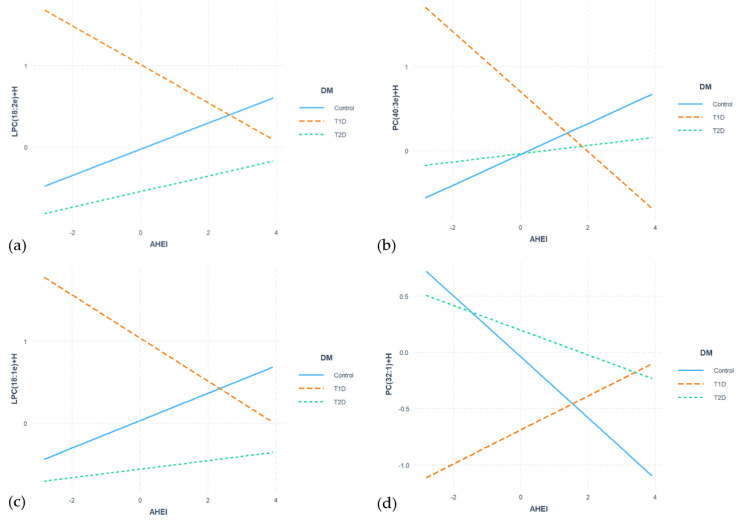
Plot of the interactions between lipids and aHEI: (**a**) Interactions of LPC(18:2e)+H; (**b**) Interactions of PC(40:3e)+H; (**c**) Interactions of LPC(18:1e)+H; (**d**) Interactions of PC(32:1)+H. aHEI, alternate Healthy Eating Index; LPC, lysophosphatidylcholines; PC, phosphatidylcholines; T1D, type 1 diabetes; T2D, type 2 diabetes.

**Table 1 nutrients-16-01805-t001:** Clinical characteristics of the study groups.

Characteristics	Control (*N* = 187)	T1D (*N* = 112)	T2D (*N* = 80)	*p*-Overall	Control vs. T1D	Control vs. T2D	T1D vs. T2D
Age (y)	54.2 (12.2)	48.2 (9.3)	58.3 (10.1)	<0.001	<0.001	0.030	<0.001
Women	87 (46.0%)	62 (52.1%)	40 (45.5%)	0.500	0.260	0.790	0.500
DM duration (y)	-	24.3 (10.5)	10.5 (7.7)	<0.001	-	-	<0.001
BMI (kg/m^2^)	26.6 (4.0)	25.9 (3.8)	31.7 (5.6)	<0.001	0.130	<0.001	<0.001
Hypertension	39 (20.6%)	38 (31.9%)	50 (56.8%)	<0.001	0.022	<0.001	0.006
Dyslipidemia	60 (31.7%)	64 (53.8%)	50 (56.8%)	<0.001	<0.001	<0.001	0.760
Smoking habit	96 (50.8%)	60 (50.4%)	54 (61.4%)	0.206	0.830	0.250	0.390
Physically active	133 (71.1%)	88 (76.5%)	40 (46.0%)	<0.001	0.460	<0.001	<0.001
Waist (cm)	96.1 (11.7)	89.7 (11.8)	106.0 (12.3)	<0.001	<0.001	<0.001	<0.001
sBP (mmHg)	125 (15.2)	130 (17.9)	140 (18.6)	<0.001	0.014	<0.001	0.014
dBP (mmHg)	78.2 (9.6)	73.6 (9.7)	76.9 (10.7)	<0.001	<0.001	0.560	0.017
HbA1c (%)	5.5 (0.4)	7.6 (0.9)	8.1 (1.5)	<0.001	<0.001	<0.001	<0.001
Total cholesterol (mg/dL)	207.0 (34.1)	182.0 (28.8)	176.0 (38.1)	<0.001	<0.001	<0.001	0.500
HDL (mg/dL)	57.9 (13.7)	65.2 (17.3)	48.8 (12.7)	<0.001	<0.001	<0.001	<0.001
LDL (mg/dL)	127.0 (29.8)	102.0 (24.3)	102.0 (33.5)	<0.001	<0.001	<0.001	0.720
TG (mg/dL)	111.0 (52.7)	72.2 (36.9)	127.0 (65.5)	<0.001	<0.001	0.030	<0.001

Data are mean (SD) for continuous variables and number (%) for categorical variables. BMI, body mass index; sBP, systolic blood pressure; dBP, diastolic blood pressure; HbA1c, glycated hemoglobin; TG, triglycerides; HDLc, high-density lipoprotein cholesterol; LDLc, low-density lipoprotein cholesterol.

**Table 2 nutrients-16-01805-t002:** Dietary intake of the study groups.

Characteristics	Control (*N* = 187)	T1D (*N* = 112)	T2D (*N* = 80)	*p*-Overall	Control vs. T1D	Control vs. T2D	T1D vs. T2D
aMED	3.3 (1.7)	3.9 (1.7)	4.2 (1.5)	<0.001	0.003	<0.001	0.230
aHEI	38.8 (7.1)	41.5 (6.4)	43.8 (6.3)	<0.001	<0.001	<0.001	0.030
Energy intake (kcal/day)	2171.0 (564.0)	2060.0 (481.0)	2165.0 (527.0)	0.212	0.100	0.840	0.220
Protein (g/day)	95.8 (15.6)	96.9 (15.9)	109.0 (24.5)	<0.001	0.420	<0.001	<0.001
Carbohydrates (g/day)	209.0 (36.8)	193.0 (33.1)	211.0 (40.3)	<0.001	<0.001	0.820	0.010
Total fat (g/day)	91.7 (15.5)	102 (15.0)	86.7 (14.0)	<0.001	<0.001	0.080	<0.001
SFA (g/day)	25.5 (5.3)	25.7 (4.7)	22.9 (4.9)	<0.001	0.860	<0.001	<0.001
MUFA (g/day)	44.2 (11.2)	52.0 (10.4)	41.3 (9.6)	<0.001	<0.001	0.180	<0.001
PUFA (g/day)	15.3 (3.7)	17.3 (5.4)	15.7 (6.3)	0.003	<0.001	0.240	0.170
Omega 3 (g/day)	1.6 (0.6)	1.7 (0.4)	1.8 (0.8)	0.002	0.070	0.002	0.080
Omega 6 (g/day)	13.6 (3.4)	15.5 (4.0)	13.8 (6.3)	0.003	<0.001	0.440	0.110

Data are mean (SD) for continuous variables and number (%) for categorical variable. aMED, alternate Mediterranean Diet score; aHEI, alternate Healthy Eating Index; MUFA, monounsaturated fat; SFA, saturated fat; PUFA, polyunsaturated fat. For the aMED and aHEI, higher scores is defined as a better adherence to the diet.

**Table 3 nutrients-16-01805-t003:** Lipids species associated with dietary intake.

m/z	Rt	Lipids	Class	q-Value	Beta	Ionization	Confirmed	Analyses
372.3107	87.1	AcCa_(14:0)+H	AcCa	0.0008	−0.29	Positive	1	aHEI_all subjects
612.5195	352.2	TG(33:1)+NH4	TG	0.0404	−0.20	Positive	0	aHEI_all subjects
878.8171	619.0	TG(52:1)+NH4	TG	0.0141	−0.23	Positive	1	aHEI_all subjects
372.3107	87.1	AcCa(14:0)+H	AcCa	0.0262	−0.31	Positive	1	aHEI_controls
640.6023	596.5	ChE(16:1)+NH4	ChE	0.0289	−0.31	Positive	1	aHEI_controls
776.5454	461.5	PC(32:1)+HCOO	PC	0.0342	−0.25	Negative	1	aHEI_controls
807.5037	436.3	PI(32:1)-H	PI	0.0308	−0.26	Negative	1	aHEI_controls
876.5768	455.1	PC(40:7)+HCOO	PC	0.0161	0.28	Negative	1	aHEI_controls
728.5193	428.4	PC(32:3)+H	PC	0.0491	0.16	Positive	0	CARB_all subjects
812.6159	493.4	PC(38:3)+H	PC	0.0171	0.19	Positive	0	CARB_all subjects
826.6306	503.3	PC(39:3)+H	PC	0.0290	0.19	Positive	0	CARB_all subjects
834.5990	461.0	PC(40:6)+H	PC	0.0316	0.19	Positive	0	CARB_all subjects
858.5996	454.1	MePC(39:5)+Na	MePC	0.0290	0.19	Positive	0	CARB_all subjects
860.6135	499.3	PC(40:4)+Na	PC	0.0019	0.24	Positive	0	CARB_all subjects
870.5999	423.6	MePC(40:6)+Na	MePC	0.0342	−0.19	Positive	0	CARB_all subjects
881.5141	438.4	PI(36:4)+Na	PI	0.0352	0.19	Positive	1	CARB_all subjects
908.7693	586.9	TG(55:7)+NH4	TG	0.0353	−0.19	Positive	1	CARB_all subjects
951.7407	584.5	TG(58:9)+Na	TG	0.0352	−0.18	Positive	1	CARB_all subjects
970.7848	580.0	TG(60:5)+NH4	TG	0.0316	−0.18	Positive	1	CARB_all subjects
733.6215	491.8	SM(d36:0)+H	SM	0.0431	−0.27	Positive	0	CARB_controls
856.6080	494.6	PC(38:3)+HCOO	PC	0.0040	0.20	Negative	1	CARB_all subjects
882.6240	500.4	PC(40:4)+HCOO	PC	0.0218	0.19	Negative	1	CARB_all subjects
710.6314	533.4	Cer(t42:1)+HCOO	Cer	0.0028	−0.32	Negative	1	CARB_controls
746.5138	468.1	CerP(m41:7)+CH3COO	CerP	0.0281	−0.27	Negative	0	CARB_controls
743.6064	472.6	SM(d37:2)+H	SM	0.0089	0.19	Positive	0	FAT all subjects
771.6375	496.4	SM(d39:2)+H	SM	0.0115	0.18	Positive	0	FAT all subjects
860.6135	499.3	PC(40:4)+Na	PC	0.0174	−0.20	Positive	0	FAT all subjects
789.6145	495.4	SM(d37:1)+HCOO	SM	0.0240	0.20	Negative	1	FAT all subjects
856.6080	494.6	PC(38:3)+HCOO	PC	0.0230	−0.19	Negative	1	FAT all subjects
882.6240	500.4	PC(40:4)+HCOO	PC	0.0346	−0.20	Negative	1	FAT all subjects
400.3417	136.9	AcCa(16:0)+H	AcCa	0.0242	−0.28	Positive	1	PROT_T2D
428.3734	215.1	AcCa(18:0)+H	AcCa	0.0421	−0.24	Positive	1	PROT_T2D
758.5689	463.8	PC(34:2)+H	PC	0.0231	0.33	Positive	0	PROT_T2D
764.5546	477.4	PC(34:3e)+Na	PC	0.0231	0.30	Positive	0	PROT_T2D
786.6003	487.6	PC(36:2)+H	PC	0.0231	0.36	Positive	0	PROT_T2D
787.6686	524.9	SM(d40:1)+H	SM	0.0231	−0.40	Positive	0	PROT_T2D
820.5249	445.8	PC(36:4)+K	PC	0.0242	0.31	Positive	0	PROT_T2D
824.5574	490.5	MePC(38:8e)+Na	MePC	0.0231	0.30	Positive	0	PROT_T2D
830.5658	460.8	PC(38:5)+Na	PC	0.0352	0.29	Positive	1	PROT_T2D
833.6497	509.2	SM(d44:6)+H	SM	0.0323	0.28	Positive	0	PROT_T2D
835.6654	528.0	SM(d42:2)+Na	SM	0.0280	0.29	Positive	1	PROT_T2D
905.7555	603.0	TG(54:4)+Na	TG	0.0291	0.28	Positive	1	PROT_T2D

*m*/*z*, mass-to-charge ratio value; rt, retention time; lipid annotation; class of the lipid; *q*-value, list of corrected *p*-values for each analysis where the lipid is considered to be significant; beta, list of linear regressors for each significant analysis; ionization, acquisition mode; confirmed, annotation via LipidSearch if they exhibited an absolute ppm error of less than 5 and an absolute retention time deviation of less than 5 s; analyses, analyses where the lipid is considered as significant. All lipids were manually annotated using MS/MS data. AcCa, acyl-carnitines; Cer, ceramides; CerP, ceramide phosphates; ChE, cholesterol esther; MePC, methyl phosphatidylcholines; PC, phosphati-dylcholines; PI, phosphatidylinositols; SM, sphingomyelins; TG, triacylglycerols. AHEI_all subjects, analysis of dietary pattern in subjects with diabetes and non-diabetic subjects; AHEI_controls, analysis of dietary pattern in non-diabetic subjects; CARB_all subjects, analysis of dietary carbohydrates in subjects with diabetes and non-diabetic subjects; CARB_controls, analysis of dietary carbohydrates in non-diabetic subjects; FAT all subjects, analysis of dietary fat in subjects with diabetes and non-diabetic subjects; PROT_T2D, analysis of dietary protein in subjects with type 2 diabetes.

## Data Availability

The data presented in this study are available on request from the corresponding author.

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
