# Peer review of "The Lipidomic Profile Is Associated with the Dietary Pattern in Subjects with and without Diabetes Mellitus from a Mediterranean Area"

_nutrients, 2024, doi:10.3390/nu16121805_

Round 1

Reviewer 1 Report

Comments and Suggestions for Authors

In the manuscript “The lipidomic profile is associated with a healthy dietary pattern in subjects with diabetes mellitus from a Mediterranean area”, Marina Idalia Rojo-López , et al. investigated important associations between lipidomic profiles and dietary habits in diabetic and non-diabetic patients in the Mediterranean region and highlighted the role of healthy dietary choices in regulating lipids.In general, in the study, the authors analyzed the relationship between lipidomics and diet in diabetic patients, with particular reference to associations with aHEI and certain lipids,Dietary carbohydrates are positively associated with several lipid types.However, there are some formatting errors in the tables and text of this article, and appropriately revisions are suggested.

Questions were raised as below and needed to be addressed.

1. The alternate Mediterranean Diet score (aMED) and the alternate Healthy Eating Index (aHEI) are not explained in the introduction, and it is difficult for the reader to understand what these two indicators tell them.

2. In result 3.3, the interaction of lipids and aMED with T2D was not characterized as statistically significant, and the study was conducted on T1D and T2D, so should the title be changed.

3. All “P-values” in the text should be formatted in italics.

4. Table 1 is incorrectly formatted, with a horizontal line missing at the end of the table.

5. The labeling of Table 3 needs to be indented by two characters in the first paragraph, in line with the previous text.

6. The table needs to be revised, the title should be in the same location, and the table header should be aligned with the content of the following

7.The “DIASTOLIC” words in line 84 should be in lowercase.

8.The “positive mode”in line 364 should be “positive ionization mode”.

Reviewer 2 Report

Comments and Suggestions for Authors

Authors identified several lipids associated with the aHEI. Some lipids had different aspects in type 1 and type 2 diabetics. This is a very interesting paper that clarifies the relationship between diet quality and lipids using a certain number of patients and appropriate controls: non-diabetic patients.

There are only a few point that is needed to be revised.

-189 Individuals without diabetes have a low prevalence of both dyslipidemia and hypertension. Why did they come to the clinic? For treatment of other diseases? Should be described in detail.

-Certain oral hypoglycemic drugs, such as SGLT2 inhibitors, are used in both type 1 and type 2 diabetes. I believe they affect the lipid profile. Are there any differences in lipids in diabetic patients depending on the oral medications they take?
